# X-ray Activated Nanoplatforms for Deep Tissue Photodynamic Therapy

**DOI:** 10.3390/nano13040673

**Published:** 2023-02-09

**Authors:** Jeffrey S. Souris, Lara Leoni, Hannah J. Zhang, Ariel Pan, Eve Tanios, Hsiu-Ming Tsai, Irina V. Balyasnikova, Marc Bissonnette, Chin-Tu Chen

**Affiliations:** 1Department of Radiology, The University of Chicago, Chicago, IL 60637, USA; 2Integrated Small Animal Imaging Research Resource, Office of Shared Research Facilities, The University of Chicago, Chicago, IL 60637, USA; 3Laboratory of Structural Biophysics and Mechanobiology, The Rockefeller University, New York, NY 10065, USA; 4Department of Neurological Surgery, Northwestern University, Chicago, IL 60611, USA; 5Department of Medicine, The University of Chicago, Chicago, IL 60637, USA

**Keywords:** photodynamic therapy, nanoparticle, X-ray activated, deep tissue, reactive oxygen species, dosimetry, photosensitizer, luminescence, scintillator, radiation therapy

## Abstract

Photodynamic therapy (PDT), the use of light to excite photosensitive molecules whose electronic relaxation drives the production of highly cytotoxic reactive oxygen species (ROS), has proven an effective means of oncotherapy. However, its application has been severely constrained to superficial tissues and those readily accessed either endoscopically or laparoscopically, due to the intrinsic scattering and absorption of photons by intervening tissues. Recent advances in the design of nanoparticle-based X-ray scintillators and photosensitizers have enabled hybridization of these moieties into single nanocomposite particles. These nanoplatforms, when irradiated with diagnostic doses and energies of X-rays, produce large quantities of ROS and permit, for the first time, non-invasive deep tissue PDT of tumors with few of the therapeutic limitations or side effects of conventional PDT. In this review we examine the underlying principles and evolution of PDT: from its initial and still dominant use of light-activated, small molecule photosensitizers that passively accumulate in tumors, to its latest development of X-ray-activated, scintillator–photosensitizer hybrid nanoplatforms that actively target cancer biomarkers. Challenges and potential remedies for the clinical translation of these hybrid nanoplatforms and X-ray PDT are also presented.

## 1. Introduction

Photodynamic therapy (PDT) has proven to be a potent, relatively non-invasive means of treating a number of oncologic pathologies that include pancreatic, esophageal, lung, and non-melanoma skin cancers [1,2,3,4,5]. Broader clinical application, however, has been limited by PDT’s operation at UV/visible/near-infrared wavelengths, where the absorption and scattering of light by tissue is appreciable and limits the depth of treatment to a few millimeters [2,6,7]. Treatment of deeper tissue tumors mandates more invasive endoscopic or laparoscopic approaches which, even then, remain limited to only a few millimeters beyond the internalized light source’s aperture, due to tissue optics [7,8,9]. Side effects of PDT such as pain, swelling, and even unintended photosensitizer activation, have also proven problematic; taking days-to-weeks to resolve, due in part to the relatively slow clearance of many photosensitizers from the body and the activation of photosensitizers by both natural and artificial light [10,11,12,13]. Photosensitizer activation by X-rays, however, precludes many of these side effects and permits non-invasive treatment of deep tissue tumors, since X-rays undergo relatively little absorption or scattering in vivo. In the following sections we review the underlying principles and evolution of PDT: from the use of optically driven, untargeted, small molecule photosensitizers to the recent development of targeted, scintillator–photosensitizer hybrid nanoplatforms that are excited by low dose, low energy X-rays.

## 2. Reactive Oxygen Species Production and Mechanisms of Action

Under homeostatic conditions, reactive oxygen species (ROSs) are normal byproducts of cell metabolism (aerobic respiration and catabolic/anabolic processes)—produced by numerous enzymatic reactions in various cell compartments that include the cytoplasm, cell membrane, endoplasmic reticulum, mitochondria, and peroxisome. ROSs are also generated within cells by specific enzymes, such as nicotinamide adenine dinucleotide phosphate (NADPH) oxidases (NOXs), and serve in signaling capacities [14]. ROSs arise naturally by exogenous means as well, such as by UV and visible wavelength photoexcitation of chromophores in exposed tissues of the skin and the eye. Under stressful conditions, however, excessive generation of ROSs trigger oxidative stress mechanisms in the cell that drive inflammation, disease, apoptosis, necrosis, and autophagy [14,15,16]. Such a paradigm is used in PDT; with systemic administration and optical excitation of exogenous photosensitive molecules that result in ROS production overwhelming the body’s antioxidant (e.g., superoxide dismutase, catalase/glutathione peroxidases, and thioredoxin) response and eliciting the aforementioned conventional forms of cell death as well as non-conventional forms such as mitotic catastrophe, pyroptosis, and parthanatos [17].

### 2.1. Molecular Radiative and Nonradiative Transitions

When a photosensitive molecule absorbs a photon, typically in the range of 600–850 nm for conventional PDT, one of its ground singlet state (S_0_) electrons is transiently excited from its low-energy molecular orbital to a higher-energy molecular orbital without changing spin, as shown in Figure 1 [18,19]. Promotion to the non-equilibrium, excited singlet state (S_1,2,…,n_) is rapid, with photon absorption taking ~10^−15^ s. Direct excitation from ground singlet state to triplet excited states (T_1,2,…,n_) is forbidden by conservation of spin angular momentum. Within ~10^−9^ s the excited singlet state decays to its ground singlet state via radiative and/or non-radiative mechanisms. First to occur (10^−12^–10^−10^ s) is non-radiative vibrational relaxation to the lowest vibrational level of the same electronic state, either by intermolecular or intramolecular energy transfer. A molecule in a higher singlet electronic state can also undergo internal conversion—a non-radiative transition between 2 electronic states of the same spin multiplicity—to a lower singlet electronic state in 10^−11^–10^−9^ s, followed immediately by vibrational relaxation to the lowest vibrational energy level of the electronic state.

The combination of vibrational relaxation and internal conversion often leave the molecule in its lowest singlet excited state (S_1_). Radiative transition from S_1_ to S_0_ then proceeds by fluorescence on timescales of 10^−10^ to 10^−7^ s, maintaining the spin multiplicity of the initial electronic state. An alternative, non-radiative, vibrational-isoenergetic transition between electronic states of different spin multiplicity can also occur. Termed intersystem crossing, this process results from spin-orbit angular momentum coupling and enables S_1_ to T_1_ transition (10^−10^–10^−6^ s), thereby circumventing fluorescence and internal conversion. Decay from the excited triplet state then proceeds via either phosphorescence (10^−6^–10 s) or vibrational relaxation. For most organic molecules, however, intersystem crossing is too slow to compete with other S_1_ decay mechanisms (fluorescence and intersystem crossing). As the first excited triplet state is much longer lived than the first excited singlet state, collisional transfer of energy to surrounding moieties such as oxygen molecules is much more probable for a molecule in the T_1_ state than in the S_1_ state. Accordingly, the majority of chemically reactive species arise while the molecule is in its T_1_ state.

### 2.2. Type I and Type II Photochemical Reactions

In photodynamic therapy, a photosensitizer molecule in its excited triplet state can undergo one of two types of photochemical reactions, termed Type I and Type II, as illustrated in Figure 1 [20,21]. In a population of photosensitizers, Type I and Type II reactions generally occur simultaneously with synergistic therapeutic outcomes, in proportion to the relative concentrations of oxygen and substrate locally available, as well as the nature of the photosensitizer employed. In Type I reactions, the photosensitizer interacts with a nearby molecule to transfer either a proton, to form a radical anion, or an electron, to form a radical cation. These radicals can then further react with nearby substrates such as cell/organelle membranes or with molecular oxygen to produce reactive oxygen species (ROS). This pathway generally involves superoxide anion (O_2_^−^) production via electron transfer from the excited triplet state of the photosensitizer to molecular oxygen [21,22,23]. While superoxide itself does not inflict significant oxidative damage, it can undergo dismutation with another nearby superoxide molecule in the presence of superoxide dismutase to form oxygen and hydrogen peroxide (H_2_O_2_), which are readily membrane permeable and diffusion rate-limited. Superoxide can also behave as a reducing agent by donating an electron to metallic ions, such as the O_2_^−^ reduction in ferric iron (Fe^+3^) to ferrous iron (Fe^+2^) via the Fenton reaction [21]. The reduced Fe^+2^ is readily oxidized by H_2_O_2_ to Fe^+3^, forming a hydroxyl radical (HO·) and a hydroxide ion (OH^−^) in the process. Fe^+3^ can then be reduced back to Fe^+2^ either by another H_2_O_2_ or O_2_^−^ molecule. Superoxide can also react with the hydroxyl radical to form singlet oxygen (^1^O_2_), or with nitric oxide (NO^−^) to form peroxynitrite (OONO^−^), another potent oxidizing molecule. Unlike superoxide, however, hydroxyl radicals cannot be eliminated by enzymatic reaction. Though short-lived in vivo (~10^−9^ s), HO· can damage virtually all forms of macromolecules including carbohydrates, nucleic acids, lipids, fatty acids, and amino acids, sometimes generating free radicals and free-radical chain reactions in the process.

In Type II reactions, singlet oxygen (^1^O_2_) is derived by direct transfer of energy from the photosensitizer to molecular oxygen and is the predominant reactive oxygen species produced. Singlet oxygen is highly reactive and readily interacts with a large number of biological substrates including nucleic acids (guanine especially), unsaturated lipids, and amino acids such as Trp, His, and Met [24,25]. Biological ^1^O_2_ reactions often lead to the formation of endoperoxides from [2 + 4] cycloadditions, dioxetanes from [2 + 2] cycloadditions, hydroperoxides from “ene” reactions or phenol oxidations, and sulfoxides from sulfides [26,27,28]. For example, cysteine and methionine are oxidized primarily to sulfoxides, histidine yields a thermally unstable endoperoxide, tryptophan reacts via a complex mechanism to produce N-formylkynurenine, tyrosine can undergo phenolic oxidative coupling, and phospholipids and cholesterol participate in ene-type reactions to provide lipid hydroperoxides [29,30,31]. Decomposition of peroxides results in the production of radicals that can then initiate a variety of biologically destructive chemical reactions.

Because of the high reactivity and short half-life of singlet oxygen (and hydroxyl radicals), only molecules that are proximal to the area of ^1^O_2_ production (i.e., photosensitizer localization) are directly/initially affected by PDT. The half-life of singlet oxygen in biological systems is generally <40 ns, due largely to chemical quenching. This results in the radius of action of singlet oxygen being as small as 20 nm, much smaller than the size of an average mammalian cell, and smaller than the size of most cellular organelles [32]. Even in the absence of chemical quenching (e.g., in ultrapure water), maximal diffusion distances of ^1^O_2_ cannot exceed 150 nm [10,33]. Although it is generally believed that ^1^O_2_ produced by Type II reactions is primarily responsible for the cytotoxic effect in PDT, recent studies suggest that radical species resulting from Type I mechanisms may lead to a substantially amplified PDT responses, especially in hypoxic environments [34,35,36,37].

### 2.3. PDT Mechanisms of Cell Death

Photosensitizer localization greatly influences the form and magnitude of cellular response following photosensitizer excitation. Polar photosensitizers tend to be internalized within cells either by lipid/protein-mediated transport or by endocytosis, whereas hydrophobic photosensitizers are able to rapidly diffuse through plasma membranes to gain cell entry. Once within cells, numerous substrates exist including organelle membranes of the endoplasmic reticulum, mitochondria, Golgi complex, lysosome, and plasma membrane [21,38,39,40]. The combination of high O_2_ solubility in lipids and high content of unsaturated fatty acids (especially double-bonds) of these membranes results in ^1^O_2_ reactions rates of 0.74–2.4 × 10^5^ M^−1^ s^−1^, as well as the production of oxygen-derived radicals [41]. Lipid peroxides, in the presence of trace metals, can decompose to form alkoxyl and peroxyl radicals that lead to additional free-radical chain reactions, disrupting lipid membranes and altering cell metabolism and signaling [42,43,44,45,46,47]. Despite the short in vivo half-life and diffusion distance of ^1^O_2_, photo-oxidation of protein amino acid residues generally occurs more rapidly, with reaction rates on the order of 10^7^ M^−1^ s^−1^ or greater for ^1^O_2_ reactions with cysteine, tyrosine, histidine, methionine, and tryptophan, due in part to their relative abundance [48]. Protein susceptibility to oxidation arises primarily from their double bonds and sulfurs, with disulfides reacting to form thiolsulfinates, and sulfides oxidizing to form sulfoxides.

Hydroxyl radicals are even shorter-lived than singlet oxygen in vivo, though HO· are far less selective in their activity; reacting with most amino acids at diffusion-limited rates. Hydroxyl radicals can undergo several types of reactions with amino acids, peptides, and proteins that include self-addition, electron transfer, and hydrogen abstraction [10,49,50,51]. Hydroxyl radicals damage both peptide backbones and amino acid side chains, generating a diverse array of radical protein derivatives in the process [52,53]. Peptide backbone cleavage is primarily initiated by H abstraction at the α-carbon position, followed by reaction with O_2_ to provide a peroxyl radical. The culmination of these events is the fragmentation and cleavage the protein backbone, with production of amide and carbonyl fragments, and peptide-bound hydroperoxides [54,55]. Hydroxyl radical oxidation of peptides and proteins can also generate free amino acids via reaction pathways that employ nitrogen-centered radicals, unrelated to α-carbon H abstraction backbone cleavage [49]. Metal-catalyzed oxidation of proteins is another significant pathway in the hydroxyl radical degradation of proteins, as metal ions preferentially bind particular sites of proteins, making them especially susceptible to selective damage [10,50,56,57,58].

DNA oxidative damage and breaks can be mediated either directly via single-electron oxidation of DNA or indirectly by generation of O_2_^−^, HO·, and ^1^O_2_ [44,59,60]. Single-electron oxidation occurs when a Type I photosensitizer, in its excited triplet state, abstracts an electron/hydrogen atom from a DNA base. Guanine is especially susceptible to one-electron oxidation due to its low ionization potential, forming a guanine cation radical intermediate that can either react directly with lysines, arginines, or serines in proteins and thereby cause DNA–protein cross-linking/aggregation, or undergo conversion to 8-oxoG or deprotonation into highly reactive guanine radicals (G(-H)·). ^1^O_2_ does not react with the 2-deoxyribose and thus cannot induce double-strand breaks [61]. However, DNA damage repair enzymes, initiated by ^1^O_2_-mediated oxidative stress, can result in the development of single-strand breaks [62]. PDT-induced lipid peroxidation of cell/organelle membranes results in the generation of reactive aldehydes and hydroxyalkenals that can react with DNA bases to form guanine derivatives and exocyclic DNA adducts that are highly mutagenic [63,64]. And, in contrast to singlet oxygen, hydroxyl radicals can readily cause base modifications by addition to double bonds, competing with hydrogen abstraction from the methyl group of thymine and the 2-amino group of guanine [65]. HO· abstraction of any hydrogen from 2-deoxyribose results in DNA backbone strand breaks [66,67].

## 3. Optical Photosensitizer Structure and Function

To be of clinical utility in PDT, a photosensitizer should possess high quantum yield of ROS, high photochemical stability, low dark toxicity, and selective accumulation within targeted pathologies. In addition, the PDT photosensitizer must minimally meet a number of photo-physical/chemical criteria that include significant photon absorption at wavelengths of 630–930 nm to ensure a good penetration of light in tissue, a high intersystem spin crossing probability between the excited singlet and triplet states, and a populated triplet state with energy higher than 0.98 eV (equivalent to a 1265 nm photon), the energy necessary to induce singlet oxygen formation from the ground triplet state.

### 3.1. Porphyrin and Porphyrin analogs: 1st/2nd-Generation Photosensitizers

At present, the majority of PDT photosensitizers in clinical use are cyclic tetrapyrrolic aromatic structures comprised of porphyrins and their analogs that include chlorins, bacteriochlorins, and phthalocyanines [3,68,69,70,71]. Porphyrins, whose prototypical base structure appears in Figure 2, contain 26 π-electrons, 18 of which lie in a planar, continuous macrocycle, with pyrrole α-carbon atoms connected via methine bridges [72,73]. This structure is extraordinarily stable, forming a nitrogen-rich central pocket that is ideal for metal incorporation. As such these anionic pockets serve as coordinatively unsaturated regions for charge transfer and adduct ligation; with reversible changes in electronic configuration (e.g., oxidation or spin states) and a large variety of moieties that can be attached at pyrrolic β sites or linked to the meso-positions at methine bridges. Although porphyrins possess relatively high quantum yields for ROS production when excited at visible wavelengths, their general susceptibility to photobleaching, low absorption in the NIR/IR (where tissues are most optically transmissive), hydrophobicity, tendency for aggregate stacking, and poor selectivity significantly limit their clinical utility. Hematoporphyrin derivative (HpD) and porfimer sodium (Photofrin) are among the most widely used first-generation photosensitizers, due to their longer wavelength Q-band absorption peak and substantial quantum yield for singlet oxygen production [3,74]. However, photon absorption and scattering by tissues at 630 nm photons limits these porphyrins to the treatment of superficial (less than 3 mm deep) pathologies, or those that can be proximally accessed either endoscopically or laparoscopically.

To address the limitations encountered with clinical application of 1st-generation photosensitizers, porphyrin bases were modified to include (a) replacement of the α, β, and other carbon atoms by nitrogen atoms, to alter molecular conformation and bioactivity; (b) addition of functional groups, to enhance targeting, widen theranostic impact, and minimize hydrophobicity; and (c) incorporation of first, second, or third row transition metal ions within their centers, to minimize systemic toxicity, alter triplet state lifetime, and shift maximal photon absorption to longer wavelengths for better photon penetration. These 2nd-generation photosensitizers, still under investigation, afford greater singlet oxygen quantum yields at wavelengths generally >650 nm, higher water solubility, lower dark toxicity, and higher tissue clearance rates than HpD and Photofrin. One group of 2nd-generation porphyrin photosensitizers garnering interest recently are hydroporphyrins [72,75].

Partial saturation of the carbon–carbon double bond between β-β positions leads to optical and photochemical properties that differ significantly from those of the base porphyrin, and give rise to intense absorption of light at longer wavelengths, as well as robust singlet oxygen production. There are 3 forms of hydroporphyrins that retain the fully conjugated, 18 π-electron aromatic structure of porphyrins: chlorins, bacteriochlorins, and isobacteriochlorins [76,77]. Chlorins have one pyrrole ring with reduced double bond, while bacteriochlorins have two pyrrole rings with reduced double bonds. These double bond reductions result in increasing photon absorption and shifting the longest wavelength Q-band absorption peaks to 650–690 nm for chlorins, and 750–790 nm for bacteriochlorins [35,68,78,79,80,81]. Among chlorins evaluated to date, three have been approved for PDT treatment: talaporfin (mono-L-aspartyl chlorin e6), Radachlorin, and temoporfin (m-THPC) [3,82,83]. Bacteriochlorins have proven somewhat problematic to synthesize, though several (Pd-bacteriopheophorbide, padeliporfin, and redaporfin) appear promising and are currently undergoing early stage clinical trials [3,84,85,86]. As with porphyrins, however, hydrophobicity remains a significant impediment to clinical adoption of many hydroporphryins. Additional examples of second-generation porphyrin/porphyrin-like photosensitizers include benzoporphyrins, texaphyrins, protoporphyrin IX (PpIX), and 5-aminolevulinic acid (ALA: biological precursor to PpIX) [3,71,87,88].

Phthalocyanines are another porphyrin-like 2nd-generation photosensitizer that exhibit red-shifted photon absorption [38]. Central pocket coordination of single metal atoms—typically Zn, Al, or Si—results in long excited triplet state lifetimes and high ^1^O_2_ generation quantum yields [89,90]. Historically, however, phthalocyanines have often been observed to undergo strong self-aggregation in aqueous environments and slow in vivo clearance. Aggregation generally proceeds along one of 2 paths that depend upon the relative alignment of the transition dipole moments on adjacent molecules, to yield H-aggregates and/or J-aggregates. In H-aggregates molecules stack predominantly face-to-face (large contact area, with strong π-π interactions), while J-aggregates form when molecules primarily stack in a head-to-tail arrangement (low area contact, with weak π-π interactions) [91]. Formation of these aggregates has important consequences for excited state energies and oscillator strengths of ground-to-excited state transitions; with strong modification of photon absorption and singlet oxygen production. Recent efforts to mitigate phthalocyanine aggregation, largely directed at minimizing H-aggregation, have led to the clinical approval of H_4_AlPCS_4_ (AlPCS_4_: aluminum (III) chloride phthalocyanine tetrasulfonate), as well as the development of 3rd-generation organic-inorganic hybrid nanoparticles based upon AlPCS_4_ (e.g., Gd_4_^3+^[AlPCS_4_]_3_^4−^) that demonstrate very high photosensitizer content, strong cellular uptake, excellent photostability, and very high singlet oxygen generation [90,92,93].

### 3.2. Nanoplatform/Hybrid 3rd-Generation Photosensitizers

Most recently, a wide variety of 3rd-generation photosensitizers have emerged to address issues of poor pathology targeting specificity, high hydrophobicity, and significant dark toxicity. These photosensitizers frequently employ 1st- and more often 2nd-generation photosensitizers as their photoactive moieties, conjugated directly to pathology-targeted antibodies/peptides/aptamers or conveyed by similarly targeted nanoplatforms that include liposomes, polymeric micelles, and a diverse array of organic/inorganic nanoparticles (e.g., polymers/dendrimers, graphenes/fullerenes/nanotubes, SiO_2_/Au/Fe nanoparticles, upconverting nanoparticles, and metal–organic frameworks), as shown in Figure 3 [94,95,96,97,98,99,100]. Liposomes and micelles were the first nanosystems to be evaluated for photosensitizer transport in PDT, with liposomes incorporating of hydrophobic agents in their lipid bilayers and/or hydrophilic agents in their aqueous cores, and block copolymer micelles encapsulating hydrophobic agents within their hydrophobic cores (Amphiphilic surfactant-based micelles are rarely used for photosensitizer conveyance owning to their low in vivo stability when compared with polymeric micelles).

To enhance targeting specificity and minimize off-target side effects such as systemic allergic reactions and skin photosensitivity, the most effective 3rd-generation photosensitizers rely upon active targeting and environmentally protected delivery of their photoactive agents. Following intravenous administration, nanoscale photosensitizers nonspecifically distribute throughout the body, with significant retention/excretion by the reticuloendothelial system (RES: primarily liver and spleen) and kidneys, despite the common use of polyethylene glycol (PEG) encapsulation of nanocarriers to minimize opsonin recognition and RES uptake. Nanoparticles that remain in circulation have the opportunity to accumulate within tumors and extravasate into the tumor’s interstitial spaces via the tumor’s intrinsically leaky vascular bed and poor lymphatic drainage. This phenomenon, known as the enhanced permeation and retention (EPR) effect, is inherently inefficient (cell uptake <1% of injected nanoparticle dose), and generally operates most efficiently for 60–200 nm diameter nanoparticles (physicochemical considerations aside), as nanoparticles smaller than this are more susceptible to diffusion, convection, and interstitial pressure gradients (i.e., metabolic clearance) while larger nanoparticles are impeded in their transport to tumor cells by the tumor’s comparatively dense, random extravascular matrix of glycosamine glycans, collagen fibers, and proteinaceous debris [101,102]. Consequently, nearly all 3rd-generation photosensitizers are designed to supplant passive targeting with active targeting, by modifying their surfaces with pathology-specific ligands and tailoring their physiochemical properties (e.g., size/shape/surface charge) and functionalities (e.g., response to internal/external stimuli such as pH, enzyme activity, redox state, temperature, light, electromagnetic fields) to promote efficient extravascular transport and tumor cell incorporation [103,104].

### 3.3. Cellular Uptake of PDT Photosensitizers

The short lifetime and diffusion kinetics of most reactive oxygen species generally limits their range of interaction to less than 20 nm and mandates cellular localization and internalization of photosensitizers prior to their photoactivation for greatest therapeutic effect, though other tumor infrastructures are also targeted for PDT, such as tumor stroma for permeabilization and vascular bed for disruption. Most nanoplatform uptake by tumor cells, whether targeted or untargeted, is size-dependent and takes place via clathrin- and caveolae-mediated endocytosis. Actin-dependent phagocytosis—via macrophages, dendritic cells, and neutrophils—is largely relegated to internalization of larger moieties (>500 μm in diameter), which can include nanoparticle aggregates and agglomerates (Figure 4) [105,106]. Non-specific internalization of nanocarriers into tumor cells via macropinocytosis and pinocytosis occurs as well, but to a lesser degree. Size-dependent rates of nanoparticle uptake have been observed to arise from competing internalization processes (e.g., receptor-mediated endocytosis of mono-disperse, individual SiO_2_ nanoparticles vs. macropinctocytosis of aggregates of the same SiO_2_ nanoparticles), as well as from single internalization processes (e.g., receptor-mediated endocytosis of differing radii, individual SiO_2_ nanoparticles) [105,106,107]. Nanoparticle shape can also dramatically affect nanoparticle uptake route and efficacy in cells [106]. For example, a variety of similar-sized gold nanoparticles in the shape of stars, rods, and triangles have been found to undergo clathrin-mediated endocytosis. However, only those configured as nanorods were also internalized via caveolae/lipid raft-mediated endocytosis, while those shaped as nanotriangles were also internalized via phagocytosis/macropinctocytosis [100].

Nanoplatform surface charge and chemistry can likewise have an enormous impact on the route and efficiency of their cellular uptake [106,108]. For example, carboxymethyl chitosan-grafted nanoparticles (negatively charged) and chitosan hydrochloride-grafted nanoparticles (positively charged) were used to evaluate the impact of surface charge on uptake efficiency by macrophages [106]. Positively charged moieties exhibited much higher phagocytic uptake than the negatively or neutrally charged moieties. And both positively and negatively charged nanoplatforms demonstrated greater macrophage uptake than neutral or PEGylated versions of the same nanoparticle. In general, net positive nanoparticle charge appears to improve the efficacy of their cellular internalization, although frequently with higher cytotoxicity [108]. Cationic nanoparticles have been found to cause more pronounced disruption of plasma membrane integrity, stronger mitochondrial and lysosomal damage, and a higher number of autophagosomes than their anionic counterparts [106]. Non-phagocytic cells ingest cationic nanoparticles to a greater extent, but surface charge density and hydrophobicity seem equally important; phagocytic cells preferentially take up anionic moieties. Surface functionalization with PEG or poloxamers (block copolymers of PEG and polypropylene glycol (PPG)) generally inhibit phagocytosis via protecting the underlying nanoparticle from ionic strength, promoting particle dispersion, and reducing surface absorption of serum proteins.

## 4. X-ray Scintillator Structure and Function

Despite the evolution of photosensitizers towards ROS generation at near-infrared (NIR) wavelengths, photon absorption and scattering limit optical PDT to treatment depths of less than ~10 mm [2,9,39]. To address this impediment, optical approaches aimed not at mitigating but circumventing photon absorption and scattering have been developed, most directly by incorporating chemiluminescent or bioluminescent sources within photosensitizer nanoplatforms, to provide unperturbed optical excitation of, or resonant energy transfer to, photoreactive centers nearby. These photosensitizers, however, have largely proven problematic to control, due to their intrinsically autonomous nature and general inaccessibility following systemic administration.

X-rays, by contrast, suffer comparatively little photon absorption or scattering as they traverse tissue, especially at diagnostic to therapeutic energies (25 keV to 25 MeV). Consequently, there has been considerable interest in developing photosensitizers that are activated by X-rays; enabling treatment of deep tumors while obviating invasive laparoscopic/endoscopic excitation, precluding non-clinical ROS generation from sources such as sunlight, and eliminating patient photosensitivity [109,110,111,112,113,114,115]. As conventional photosensitizers cannot directly absorb such energetic photons, wide-bandgap scintillators are often co-incorporated proximal to photoreactive centers to serve as wavelength shifters that absorb and convert a portion of the incident X-ray’s energy to light of suitable wavelength for photosensitizer excitation.

### 4.1. Mechanism of Scintillation

When X-rays enter a scintillator, photon energy loss takes place by three energy-dependent mechanisms: photoelectric absorption, Compton scattering, and electron-positron pair production. For lower energies (up to a few hundred keV), photoelectric absorption dominates. In this process, photon energy transfer to scintillator atoms results in the ejection of photoelectrons whose kinetic energy corresponds to the difference between the electron’s binding energy and the photon’s incident energy. This is followed by a redistribution of electrons to fill the photoelectron’s vacancy, resulting in the production of Auger electrons and emission of characteristic X-rays. At energies greater than a few hundred keV, incident photons transfer a portion of their initial energy to the scintillator’s atomic electrons, resulting in the inelastic scattering of photons and recoil of electrons from which they scattered; a process known as Compton scattering. The recoil electron’s energy, corresponding to the difference between the incident photon’s energy and the sum of the target electron’s binding energy and scattered photon’s energy, is then rapidly reabsorbed by the scintillator in a continuum fashion up to the incident photon’s energy less one-half the electron’s rest mass. As such, highly energetic free electrons can be generated that, in turn, go on to ionize or excite additional scintillator atoms in cascade. For X-ray energies that exceed twice the electron’s rest mass (i.e., >1.02 MeV), electron-positron pair production arises from energetic photon interaction with scintillator nuclei (and to a lesser degree, scintillator electron) electric fields.

The scintillator’s elemental composition and internal structure principally determine the efficiency with which an incident X-ray’s energy is down-converted to scintillation light. Organic scintillators are largely comprised of low atomic number (low-Z) elements and thus typically have limited photoelectric interactions, though introduction of high-Z elements such as the iodine and bromine, can significantly enhance their X-ray absorption cross-section, intersystem crossing (triplet generation), and radioluminescence production. Inorganic scintillators, by contrast, are generally comprised of high-Z elements and possess much higher densities, making photoelectric interactions significantly more probable. In these inorganic, structured materials, incident electrons in the keV range couple with atomic lattice electrons and excite the latter from occupied valence and inner core bands to different energy levels within the conduction band, as shown in Figure 5. Each of these excitations generates an electron–hole pair, with hole “depth” and electron kinetic energy reflecting the energy imparted by the incident electron. Within a very short period (10^−16^–10^−14^ s), inelastic electron–electron scattering and short-range Auger processes cause a multiplicative cascade of secondary electronic excitations, further populating the conduction band with electrons, and the valence and core bands with holes. 

Provided the electron’s energy is sufficient to achieve ionization threshold, free carriers are produced that move randomly through the scintillator lattice until they are trapped by endogenous/exogenous defects or recombine at luminescent centers. If the electron’s energy, however, is insufficient to achieve ionization threshold, the electron–hole pair releases a portion of their energy to the lattice’s vibration modes by phonon coupling until the low kinetic energy electron occupies the bottom of the conduction band and its corresponding hole occupies the top of the valence band (or they bind to form an exciton whose energy is slightly less than the valence-conduction bandgap). This thermalization process takes approximately 10^−14^ to 10^−12^ s to occur. During the next 10^−12^ to 10^−10^ s, excitations localize with stable lattice defects, impurities, and luminescent centers, with sequential charge carrier capture or other energy transfer mechanism exciting luminescent centers in 10^−10^ to 10^−8^ s. In the final step, the excited luminescent center returns to the ground state by photon emission or nonradiative suppression. The radiative process is comparatively slow (10^−9^ to 10^−3^ s) for electron–hole pair recombination, exciton emission, or electronic recombination, though it can take up to several minutes to occur if the process is a highly prohibitive state [116].
Figure 5Schematic of general scintillation mechanism and density of states of an ionic crystal scintillator: e: electrons; h: holes; ph: phonons; hn: photons; V_k_: self-trapped holes; STE: self-trapped excitons; c^n^: ionic centers of charge n and excitation status *; grey/white regions denote, respectively, electron/hole density of states. Reproduced with permission [117] and A.N. Vasil’ev. Copyright 2020, Springer.
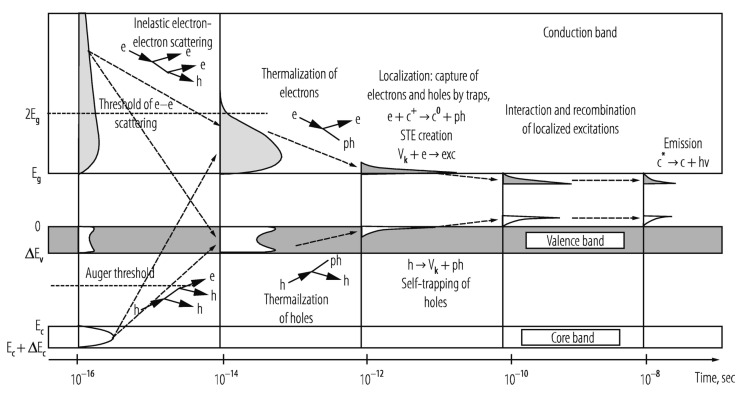


### 4.2. Nanoscale Form Factor Implications

While the preceding description of scintillation processes applies to both bulk and nanoscale scintillators, there are differences between the two form factors [118,119]. For bulk scintillators, elemental composition and structure are the primary determinants of luminescence efficiency. Photoabsorption cross-sections of inner-shell electrons are much larger than those of the outer-shell electrons. Consequently, scintillators containing high-Z elements, which have greater numbers of inner shell electrons, tend to be more efficient at capturing X-rays than low-Z materials. And, as such, the majority of bulk scintillators consist of high-Z crystalline matrices, doped with rare earth elements. Luminescence yield is further enhanced by minimizing the number of lattice defects within the bulk scintillators during synthesis (e.g., annealing), since these are sources of nonradiative processes that directly compete with, and thereby reduce, the emissions of luminescence centers. In many bulk crystalline scintillators, annealing can result in exponential increases in radioluminescence emission intensity [120].

For nanoparticle scintillators, high-Z elements are employed as well, just as in bulk scintillators. However, the inherently high surface area to volume ratio of nanoscintillators results in a significant fraction of scintillator activator sites residing on the nanoparticle’s surface rather than deeply within. This effective reduction in system dimensionality, and the intrinsically high mobility of excitations induced, leads to more frequent stimulation of scintillation activator sites in nanoscintillators than in their bulk counterparts [113,121]. The enormous surface area to volume ratio of nanoparticle scintillators has other consequences as well. In bulk scintillators, electron–hole pairs lose energy via phonon coupling to the lattice, while in nanoscale scintillators spatial confinement results in energy quantization levels that impede phonon emission; with energy transfer that instead leads to the creation of new electron–hole pairs, increasing the probability of recombination at luminescence centers [119]. Likewise, lattice pressure gradients, acquired during nanoparticle synthesis, often lead to crystal field surface fluctuations that tend to randomize the distribution of energy levels, decreasing the probability of direct energy transfer and, consequently, increasing the probability of radiative transition [113]. Taken together, nanoparticle scintillators generally have greater luminescence efficiency, per unit mass, than bulk scintillators of identical elemental composition. Even semiconductors and pure metals, which do not generally function as X-ray scintillators in bulk form, can function as such if they are restricted to nanoscale dimensions, due to surface effects and quantum confinement [113,119]. 

The nanoscintillator’s enormous surface area and diminutive volume have significant implications for in vivo functionality as well. While the expansive surface provides abundant sites for the conjugation of targeting ligands and other functional moieties, surface luminescence can also be exquisitely sensitive to local environmental conditions unless protected. Changes in the local milieu’s pH and ionic strength, and the adsorption of biochemical/molecular species can adversely affect excitation lifetime/propagation and luminescence efficiency. The small volume fraction occupied by nanoscintillators in vivo (typical physiological concentrations are less than 1 mg/mL) results in most X-ray energy deposition occurring not within the high-Z nanoparticle itself, but within the surrounding water molecules, driven by inelastic electron scattering. Indeed, even secondary electrons generated within the nanoscintillator have ranges that are much greater than the size of the nanoscintillator [114,122]. Therefore, a nanoplatform’s electron absorption cross-section more closely reflects its X-ray PDT efficacy than its X-ray absorption cross-section, unless very high local concentrations (non-aggregated/agglomerated) of nanoscintillators are obtained in vivo.

## 5. X-ray Photosensitizer Structure and Function

For clinical application in X-ray PDT, nanoscale photosensitizers must meet a number of physical and chemical criteria. Scintillators should possess large X-ray and electron absorption cross-sections, bright radioluminescence (more than 15 photons/keV), high irradiation stability, and good biocompatibility. Photosensitizers should have excellent irradiation stability, significant spectral overlap between their photoabsorption band and the scintillator’s photoemission band, long excited state lifetime, high quantum yield of ROS, and close proximity (less than 10 nm) to scintillator luminescence centers, to maximize optical and resonant energy transfer. Scintillator–photosensitizer nanoplatforms should also be surface-functionalized for enhanced biodistribution/biocompatibility and pathology targeting, and protected from environmental factors that can degrade radioluminescence and ROS generation efficiency, such as variations in pH, ionic strength, and adsorption of biomolecules. Since the performance of photosensitizers are also highly sensitive to their local microenvironment (e.g., pH, O_2_, ionic strength, molecular adsorption), the manner in which they are incorporated within the nanoscintillator matrix is of great importance. The four most commonly used methods of nano-photosensitizer/scintillator association are pore loading, electrostatic/hydrophobic interaction, covalent conjugation, and direct coating/incorporation, with covalent bonding and direct coating/incorporation generally proving the most stable, and thus the best suited, for in vivo use.

### 5.1. Structural/Functional Hybridization

The number of scintillator–photosensitizer permutations that have been developed and evaluated is enormous and rapidly expanding, though only a few have entered into investigational clinical trials to date. In the broadest sense, most of these X-ray PDT nanoplatforms can be categorized as belonging to one of 4 groups that are based on scintillator material: rare earth elements, metals, quantum dots, and silicon. Representative forms of these nanoplatforms are shown schematically in Figure 6 and Figure 7 [123]. In some X-ray PDT implementations, scintillators and photosensitizers are delivered independently of one another. An example of this approach, currently undergoing Phase I clinical trial (NCT04389281), is X-ray Psoralen Activated Cancer Therapy (X-PACT). In this technique, X-ray/UV down-converting rare earth nanoscintillators such as Y_2_O_3_ or nanophosphors such as CaWO_4_ are co-administered with UV-activated psoralen-derivative photosensitizers such as 8-Methoxypsoralen (8-MOP), to yield deep tissue apoptosis upon X-ray irradiation [124,125]. More often than not, however, spatially optimized combinations of scintillator and photosensitizer are used within the same nanoplatform (e.g., high-Z metal inclusion, for enhanced X-ray absorption, in lanthanide-doped oxides and semiconductors), to synergistically exploit the strengths of each isoform and yield a more promising clinical candidate, via mechanisms of action that are not always readily apparent and potentially multifactorial.

Doped rare earth elements were among the first nanoplatforms to be explored for use as X-ray-induced scintillators in the PDT of cancer, and remain among the most prolifically studied [139,140]. These early studies employed fluorides and oxides of scintillators (e.g., LaF_3_:Ce^3+^, LuF_3_:Ce^3+^, Tb_2_O_3_, and BaFBr:Eu^2+^) as transducers with spectrally matched organic photosensitizers (e.g., porphyrin, protoporphyrin IX, and Rose Bengal), due to appreciable spectra overlap between the scintillator’s emission and the photosensitizer’s absorption bands. Porphyrins were chosen both because of their approved clinical use as conventional photosensitizers in PDT and their multiband absorption spectra, which includes a dominate Soret band near 400 nm as well as longer wavelength Q-bands that enhance their quantum efficiency. For example, Tb-containing platforms typical exhibit emission spectrums comprised of peaks centered at 488 nm (^4^F_8_—^4^F_8_ ^5^D_4_ → ^7^F_6_ transition), 545 nm (^5^D_4_ → ^7^F_5_ transition), 588 nm (^5^D_4_ → ^7^F_4_ transition), and 625 nm (^5^D_4_ → ^7^F_3_ transition).

The concept of X-ray-induced PDT using scintillating nanoparticles was first proposed by Chen et al. [139] in 2006, with the earliest studies employing Tb-based fluoride/oxide crystalline nanoparticles such as LaF_3_:Tb/Ce or Tb_2_O_3_ as X-ray energy acceptors. Surface conjugation of conventional photosensitizers such as porphyrin and porphyrin derivatives to these nanoparticles generated several times greater quantities of ^1^O_2_ under X-ray irradiation than can be generated via the photosensitizers alone [140]. Subsequent studies by the same group, involving the conjugation of meso-tetra(4-carboxyphenyl) porphine to the surface of LaF_3_:Tb nanoparticles, resulted in X-ray activation at lower X-ray dose and dose rate (250 keV, 0.44 G/min for 30 min) than previously, with little loss of singlet oxygen generation efficiency even when further functionalized for cancer targeting by folic acid conjugation [141]. However, the quantum efficiency of ROS generation of these X-ray activated nanoplatforms systems was significantly less than that of optically activated nanoplatforms bearing conventional photosensitizers.

### 5.2. Illustrative Recent Advances in X-ray Photosensitizer Design

To further improve the quantum efficiency of ROS generation in X-ray PDT, Tang et. al. [127] used the same nanomaterial, but synthesized via a facile 2-step hydrothermal process without surfactant, template, or catalyst. This approach resulted in the production of monodisperse mesoporous LaF_3_:Tb crystalline nanoparticles that possessed readily accessed, but highly heterogeneous, pore structure, as schematically shown in Figure 8.

As the pores directly communicated with the surrounding microenvironment, aqueous co-suspension of the nanoparticles with a water-soluble photosensitizer (Rose Bengal) enabled simple/passive pore loading. The final nanoplatform’s greatly reduced separation between Tb^+3^ activators in crystal and Rose Bengal in pores, significantly enhanced Förster resonance energy transfer (FRET) between the moiety’s scintillator and photosensitizer. Energy transfer efficiency was determined to be as high as 85% due to the near exact overlap of LaF_3_:Tb’s emission band at 544 nm and Rose Bengal’s primary absorption band at 549 nm. Post X-ray irradiation singlet oxygen measurements of LaF_3_:Tb-RB revealed significant ^1^O_2_ generation relative to nanoparticle-free Rose Bengal suspensions of the same fluorophore concentration.

While Tb^+3^ doping of oxides often results in green luminescence, Eu^+3^ doping of the same compounds generally shifts luminescence to longer, red wavelengths. Europium-doped yttrium oxide (Y_2_O_3_:Eu) phosphors are widely known to produce intense red emission under short wavelength (210–255 nm) UV excitation, corresponding to the charge transfer from O^2−^ → Eu^+3^ that promotes electronic transition between O_2_-2p orbital and the unfilled Eu^+3^-4f orbital [142]. Bulk Y_2_O_3_:Eu films that scintillate under X-ray irradiation have previously been evaluated for use as X-ray detectors in medical imaging, due to their significant radioluminescence at clinically relevant X-ray energies and fluences [143]. In 2014 Souris et al. [144] synthesized and characterized annealed Y_2_O_3_:Eu nanoparticle clusters with the aim of developing quantitative, cancer-targeted nanodosimeters to be used in conjunction with external beam radiation therapy, as shown in Figure 9. Spectroscopic analyses of these nanoparticles during X-ray irradiation revealed surprisingly bright and stable radioluminescence at near-infrared wavelengths that were amenable to both in situ surface radiance measurements and diffuse optical tomography, with markedly linear response to changes in X-ray flux and energy. Monte Carlo modeling of incident flux and broadband wide-field imaging of mouse phantoms bearing both Y_2_O_3_:Eu nanoparticles and calibrated LEDs of similar spectral emission, demonstrated significant radioluminescence of high quantum efficiency, in agreement with quantitative imaging and spectroscopic studies [145]. Subsequent in vivo studies of Y_2_O_3_:Eu@SiO_2_ nanoparticles by the same group (Figure 9e,f, unpublished) using [^18^F]FLT (fluorothymidine), a cell proliferation marker for PET imaging, revealed diminished [^18^F]FLT uptake and tumor volume in mice bearing human ovarian cancer (Caov3) xenografts that received Y_2_O_3_:Eu@SiO_2_ X-ray PDT versus those that did not receive nanoparticle administration [146]. More recent efforts have focused on in vivo studies using a novel monoclonal antibody (mAb47) to target these nanoplatforms to IL13RA2 overexpression in ovarian and metastatic colorectal cancer [147]. In 2020 Chuang et al. [109] investigated similar Y_2_O_3_:Eu nanoscintillator clusters encased within ~10 nm thick SiO_2_ shells, for use in X-ray PDT, demonstrating the substantial production of superoxide, hydroxyl radical, and singlet oxygen under modest X-ray irradiation energies and intensities, without the presence of photosensitizers, as shown in Figure 9. In these studies, a portion of the X-ray’s energy was postulated to be transferred to Eu^+3^ for radioluminescence production while another portion was directly converted into the generation of electron–hole pairs that, when trapped on the nanoparticle’s surface, reacted with aqueous electron acceptors (i.e., molecular oxygen) and donors (i.e., water and hydroxyl ions), respectively, to produce different forms of ROS (i.e., electrons driving superoxide anion production, holes driving hydroxyl and singlet oxygen production) via both Type I and II processes. Clonogenic assays conducted with radiation-sensitive Caov3 and radiation-resistant SKOV3 human ovarian cancer cells revealed 40–50% therapeutic enhancement afforded by the Y_2_O_3_:Eu@SiO_2_ nanoparticles over radiation alone. Fractionated radiation treatments of nude mice bearing subcutaneous SKOV3 tumors also exhibited significant delays in tumor growth, as shown in Figure 9j.

Numerous other nanoplatforms have also been developed for direct generation ROS under X-ray irradiation without use of photosensitizers. In 2007, Takahashi and Misawa [148] examined direct X-ray-induced ROS generation in aqueous suspensions of a diverse array of nano/microparticles (TiO_2_, ZnS:Ag, CeF_3_, and CdSe quantum dot), finding significant superoxide anion and hydroxyl production at diagnostic X-ray energies and doses. In 2014, Ma et al. [149] synthesized copper-cysteamine complex (Cu–Cy) nanoparticles (Cu_3_Cl(SR)_2_; R = CH_2_CH_2_NH_2_) that produced ^1^O_2_ and bright radioluminescence by both X-ray irradiation and UV light (365 nm) exposure. With ROS production comparable (under UV excitation) or superior (under X-ray activation) to that of PPIX using UV excitation, these Cu-Cy nanoparticles demonstrated significant PDT effect in both in vitro and in vivo studies, though cellular uptake was sub-optimal: a phenomenon they attributed to their nanoparticle’s lack of cell targeting and large size (~250 nm diameter), both adversely affecting internalization. Kirakci et al. [150] in 2016 synthesized a novel octahedral molybdenum cluster compound (n-Bu_4_N)_2_[Mo_6_I_8_(OOC-1-adamantane)_6_] that generated identical luminescence spectra whether irradiated by UV light or X-rays (i.e., via same excited triplet states) that is quenched by molecular oxygen to generate singlet oxygen with high (0.76) quantum efficiency. Sulfonated polystyrene nanofibers were incorporated to enhance energy transfer and oxygen diffusion, and enable aqueous dispersion. Others have incorporated photosensitizers for ROS generation from X-ray absorbers, but without optical excitation; relying, instead, upon Förster resonance energy transfer between the moiety’s scintillator and photosensitizer that approaches 100% efficiency. Yang et al. [135] in 2008 conjugated Photofrin to PEGylated amine-functionalized quantum dots in a 291:1 ratio, to derive near 100% resonance energy transfer. With completely quenched fluorescence under 6 MV X-ray irradiation, the construct exhibited potent cytotoxicity in clonogenic assays of H460 cells.

To synchronize radiotherapy with PDT of deep hypoxic tissues, Zhang et al. [129] in 2015 combined a nanoscintillator with a semiconductor for energy down-conversion. This construct was comprised of a core–shell LiYF_4_:Ce^3+^@SiO_2_@ZnO-PEG nanoparticle (SZNP) that was synthesized from an octahedral Ce^3+^-doped LiYF_4_ nanoscintillator that was then successively enveloped within SiO_2_ followed by thiol-group grafting of ultra-small ZnO semiconductor nanoparticles onto the LiYF_4_:Ce^3+^@SiO_2_ mother nanoparticle. The entire nanoparticle was then PEGylated to enhance the overall biocompatibility and biodistribution of the final SZNP, as shown in Figure 10.

These spherical core–shell SZNPs possessed an average diameter of 33.8 nm with very high polydispersity. Respective absorption and emission spectra of LiYF_4_:Ce^3+^ and ZnO under X-ray excitation, showed excellent overlap that enabled efficient resonance energy transfer between the LiYF_4_:Ce^3+^ core and the surrounding ZnO nanoparticles. Efficient energy transfer was also supported by the observed quenching of the fluorescence of the SZNP, providing additional evidence that the SiO_2_ interlayer was not absorbing light. Nanoparticle X-ray irradiation studies in water were conducted using 0–20 Gy doses and revealed a decline in the oxygen dependency of the ROS yield through the production of O_2_^•−^ radicals involving water as the ROS source. In operation, X-ray excited LiYF_4_:Ce^3+^ cores are postulated to emit photons of low energy that match the bandgap of surface-bound ZnO nanoparticles. The subsequent excitons (electron–hole pairs) then interact with water and oxygen molecules to form free radicals. In analogy to Type I PDT processes, highly reactive hydroxyl radicals are derived from the interactions between the holes and the absorbed water instead of O_2_, which essentially minimizes the oxygen-tension dependency for the generation of reactive oxygen species. In vitro and in vivo studies on the impact of the SZNPs in HeLa cells, under either normoxic (21% O_2_) or hypoxic (2% O_2_) conditions and undergoing X-ray radiation (2.5 Gy/min, total of 2, 4 and 6 Gy for in vitro and 8 Gy for in vivo studies), revealed potent antitumor therapeutic efficacy.

In 2015 Chen et al. [128] reported the development of an X-ray PDT nanoplatform based upon the Type II PDT mechanism and achieved efficient, low-dose therapeutic response. SrAl_2_O_4_:Eu^2+^ (SAO) was synthesized using a carbothermal method that employed high temperature and pressure vapor-phase deposition. Following sedimentation/filtration/centrifugation, the SAO was encapsulated within two distinct layers of SiO_2_. The innermost layer of the SAO@SiO_2_ nanoparticle was comprised of solid silica and functioned as an environmentally protective shell. The outermost layer, however, was comprised of mesoporous silica and served as a surface-binding reservoir for passively loaded, merocyanine 540 (MC540) photosensitizer molecules, as shown in Figure 11.

The derived M-SAO@SiO_2_ nanoparticles ranged in size from 255 nm to 560 nm in diameter, and exhibited well-aligned spectral overlap of the SAO’s radioluminescence with the MC540’s absorption band that enhanced energy down-conversion efficiency. Interestingly, storage of the nanoparticles in simulated body fluids resulted in the dissolution of the cores within 14 days, leaving behind empty silica shells. Under X-ray irradiation, the nanoparticles produced significant quantities of ^1^O_2_, as quantitated by Singlet Oxygen Sensor Green (SOSG) assay. Biological effect of the nanoplatform was assessed via in vivo and in vitro studies employing human glioblastoma U87MG cells and murine tumors. Compared with the observed low cytotoxicity in comparably treated controls, X-ray irradiation of radioresistant U87MG cells pre-incubated with M-SAO@SiO_2_ showed viability drops to 38%. Additionally, in vivo X-ray PDT studies demonstrated that subcutaneous U87MG tumor growth was almost completely inhibited following intratumoral injection of the M-SAO@SiO_2_ nanoparticles, compared with the controls.

In 2015 Rossi et al. [136] reported the growth of inorganic core–shell SiC/SiO_x_ (1.8 < x < 2) nanowires (NWs) on silicon substrates in a chemical vapor deposition system using a vapor-liquid-solid process that was catalyzed by iron. The resulting nanoplatform exhibited an optical emission spectrum that matched well with the absorption bands of many organic photosensitizers including porphyrins. The investigators selected the porphyrin tetra(4-carboxyphenyl)porphyrin (H_2_TCPP) for incorporation, with carboxy groups converted into amides containing a short chain that possessed a terminal alkyne functional group, as shown in Figure 12. 

The nanowires were then functionalized with azide groups and covalently coupled with the porphyrin derivative through click chemistry. This resulted in <10 nm separation between nanowire and photosensitizer to enable Förster resonance energy transfer between donor and acceptor. Scanning electron microscopy (SEM) imaging of the inorganic component of the nanoplatform revealed that the nanowires were arranged in dense tangles, while transmission electron microscopy (TEM) studies showed crystalline 3C-SiC cores (average diameter 20 nm) coated with amorphous SiO_x_ shells (~20 nm), with catalyst nanoparticle tips. Singlet oxygen production was evaluated using SOSG assays with 6 MV X-ray irradiations at low dose (0.4–2.0 Gy). Pre/post X-ray irradiation (0.6, 1.2, and 2.4 Gy) TEM studies were conducted to rule out amorphous-to-crystalline SiO_x_ transition since the crystalline phase is known to be cytotoxic. SOSG fluorescence, reflecting ^1^O_2_ generation, rose rapidly with dose until saturating around 1 Gy. SOSG fluorescence signal saturation at 1 Gy was attributed to the finite capacity of SOSG molecules to detect ^1^O_2_ immediately after it has been produced: limited, on one side, by the short lifetime of this species in water (~4 ms) and on the other side by the exchange/diffusion of SOSG molecules between the nanoplatform’s surface and the surrounding solution. In vitro clonogenic survival assays of human lung adenocarcinoma cells (A549) revealed that 12 days after X-ray irradiation at a dose of 2 Gy, cell populations were reduced by about 75% with respect to control cell populations. In 2017, Tatti et al. [151] described the surface functionalization of SiC/SiO_x_ core/shell nanowires with tetrakis(pentafluorophenyl)porphyrin (H_2_TPP(F)) using supersonic molecular beam deposition rather than chemical vapor deposition. This work resulted in the formation of a stable nanowire-H_2_TPPF nanoplatform due to strong interactions between the reactive fluorine atoms on the peripheral position of the organic photosensitizer and the inorganic SiO_x_ shell of the nanowire. Especially noteworthy is that significant ^1^O_2_ production was observed with these moieties upon 6 MV X-rays irradiations at 1.2 Gy doses, a high-energy down-conversion not frequently observed with nanoscintillators.

In 2017, and building upon their group’s prior work with chlorin-based 3D nanoscale metal–organic frameworks (nMOFs) [152,153,154], Lan et al. [155] described their development of novel 2D metal–organic layers (MOLs) of 1.2 nm thickness; a topology designed specifically to facilitate the diffusion of short-lived/range ROS. The MOLs were composed of Hf_6_O_4_(OH)_4_(HCO_2_)_6_ secondary building units (SBUs), with iridium Ir[bpy(ppy)_2_]^+^ or ruthenium [Ru(bpy)_3_]^2+^ complexes, where bpy denotes 2,2′-bipyridine and ppy denotes 2-phenylpyridine)—for brevity labeled Hf-BPY-Ir and Hf-BPY-Ru MOLs—as shown in Figure 13.

The iridium and ruthenium complexes BPY-Ir and BPY-Ru proved highly efficient photosensitizers, with singlet oxygen production quantum yields of 0.97 at 355 nm excitation and 0.73 at 450 nm excitation, respectively. Inductively coupled plasma mass spectroscopy (ICP-MS) of Hf-BPY-Ir and Hf-BPY-Ru MOLs revealed Ir and Ru loading efficiencies of 67% and 59%, respectively. TEM imaging showed Hf-BPY-Ir and Hf-BPY-Ru each morphologicaly similar to Hf-BPY, while powder X-ray diffraction demonstrated retention of MOL structure following Hf metalation and 12 hr DMEM media incubation. Singlet oxygen generation efficiencies of MOLs were quantified using 4-nitroso-*N*,*N*-dimethylaniline (RNO) assays. Zr-MOLs (Zr-BPY-Ir and Zr-BPY-Ru) were synthesized and evaluated similarly, for comparison. Visible light irradiation showed that Ir-based Zr- and Hf- MOLs generated ^1^O_2_ more efficiently than Ru-based Zr- and Hf- MOLs, consistent with their respective difference in ^1^O_2_ generation quantum yields. However, X-ray irradiation revealed a dramatic difference in ^1^O_2_ generation efficiencies of Zr- and Hf- MOLs: both Hf-MOLs had much higher ^1^O_2_ generation efficiency than their corresponding Zr-MOLs, supporting their hypothesis that X-ray energy was first absorbed by SBUs (heavier Hf doing so more efficiently than lighter Zr) and then transferred to the photosensitizers in the bridging ligands to lead to the singlet oxygen generation. 

In vitro anti-cancer efficacy studies of Hf-based MOLs were conducted using CT26 and MC38 murine colon adenocarcinoma cells, with Zr-MOL serving as controls. Hf-BPY-Ir and Hf-BPY-Ru outperformed Hf-BPY and all Zr-MOLs; with Hf-BPY-Ir, Hf-BPY-Ru, and Hf-BPY MOL IC_50_ values of 3.82 ± 1.80, 3.63 ± 2.75, and 24.90 ± 7.87 μM in CT26 cells and 11.66 ± 1.84, 10.72 ± 2.92, and 37.80 ± 6.57 μM in MC38 cells, respectively. By comparison, IC_50_ values exceeded 100 μM for Zr-BPY-Ir, Zr-BPY-Ru, and Zr-BPY in both CT26 and MC38 cell lines. 

In vivo anticancer efficacy studies of Hf-BPY-Ir, Hf-BPY-Ru, and Hf-BPY MOLs (Ir, Ru or BPY of 0.5 nmol) were also conducted using sub-cutaneous flank CT26 and MC38 tumors in mice. Daily fractionated X-ray dosing—1 Gy given 5 days for CT26 and 10 days for MC38—of Hf-BPY groups appeared to show slight inhibition of tumor growth, consistent with the radiosensitization effects of the Hf_6_ SBUs. However, Hf-BPY-Ir and Hf-BPY-Ru treatments led to very significant tumor volume reduction in CT26 models of 83.6 % or 77.3 %, respectively, and in MC38 models of 82.3 % or 90.1 %, respectively.

In 2020 Sun et al. [156] reported the synthesis and evaluation of photosensitizer-conjugated aggregation-induced emission (AIE) gold “clustoluminogens”—derived from the integration of glutathione-protected gold atom clusters (GCs) into larger, heterogeneous aggregates that exhibited AIE, as shown in Figure 14. TEM imaging during synthesis revealed the ultra-fine GCs to be approximately 2.5 nm in diameter. Previous studies by others had demonstrated that protein-protected ultra-fine gold atom clusters can emit substantial radioluminescence during X-ray excitation, and were potentially well-suited for use as contrast agents [157,158]. However Sun et al. [156] found their AIE-GCs to have 3.1-fold enhanced fluorescence emission under UV excitation, and 5.2-fold enhanced radioluminescence emission under X-ray excitation, compared with their non-aggregated, gold atom cluster constituents under identical excitation. Using a high molecular weight cationic polymer (PAH) suspension, GCs were observed to self-assemble into AIE-GC clustoluminogens of ~65 nm diameter. Supported by elemental mapping analysis and X-ray photoelectron spectroscopy (XPS), the investigators speculated that the observed AIE phenomenon was mediated through modification of GC cross-linking by the high molecular weight cationic polymers, resulting in greatly increased electrostatic interactions between proximal GCs and significant modification of ligand-to-metal charge transfer. However, theoretical understanding of the mechanism behind AIE has yet to be established.

Since the radioluminescence emission spectra of AEI-GCs closely matched the absorption spectra of the clinically approved photosensitizer Rose Bengal (RB), RB was then conjugated onto GCs through EDC/NHS chemistry. The resulting GC-RBs were then allowed to self-assemble into AIE-GCs using the same cationic polymer approach as before, to derive AIE-Au constructs. RGD peptide was then conjugated onto AIE-Au through EDC/NHS chemistry to derive R-AIE-Au spherical nanoplatforms approximately 68 nm in diameter. In vitro studies using 9,10-anthracene-diyl-bis (methylene) dimoalonic acid (ABDA) and Methylene Blue (MB) were then conducted to determine the generation efficiency of singlet oxygen and hydroxyl radicals as a function of X-ray irradiation dose, revealing copious production of both species. Clonogenic assays of human glioblastoma (U87MG), hepatocellular carcinoma (HepG2), and prostate cancer (PC3) cells demonstrated potent PDT effect at very low (1 Gy) X-ray doses with minimal nanosensitizer cytotoxicity, as assessed via MTT assay. In vivo studies of mice bearing tumors of the same cellular composition, demonstrated similar potent PDT response, as shown in Figure 14. Noteworthy of these preclinical studies is that nanoplatform (R-AIE-Au) administration was achieved by tail vein injection at clinically relevant doses (20 mg/kg) and thus subject to RES clearance as well as systemic, non-specific binding.

## 6. Autonomous/Internal-Light Nanoplatform PDT

When a charged particle (e.g., beta particle, positron) moves through a dielectric medium (e.g., water, serum) faster than the phase velocity of light in that medium, its passage results in the production of Cerenkov radiation (CR); visible electromagnetic radiation whose intensity varies inversely with the square of its wavelength. Over the last 2 decades, CR has emerged as a useful tool for preclinical optical imaging of internalized radionuclides, limited in the depth of its application by its dominant, shorter wavelength emissions that are subject to significant photon absorption and scattering in tissue. CR can, however, be used to directly excite fluorophores, phosphors, and quantum dots, provided the CR source is not too far removed from the optical probe.

The use of CR to activate PDT requires proximal colocalization of the radionuclide and the photosensitizer [113,159]. In one proof-of-concept study, Kotagiri et al. [160] used tumor-targeted TiO_2_ nanoparticles as photosensitizers with systemically administered PET radiotracer ^18^FDG as the positron-emitting inducer of CR. (^18^FDG tends to accumulate preferentially in tumors, due to their higher metabolic activity and deoxyglucose trapping). Titanocene, a photoinitiator, was attached to the tumor-targeted (apo-transferrin labeled) TiO_2_ nanoparticles to enhance the nanoplatform’s overall cytotoxic effect. The combined effects of the complimentary radical-generation mechanisms of photocatalysts (hydroxyl and superoxide radicals) and photoinitiators (photofragmentation) enabled an effective CR-induced phototherapy in both in vitro and in vivo preclinical studies using human fibrosarcoma (HT1080) cells and tumors. Further improvements in CR-mediated PDT might be realized with the employment of higher energy β^+/−^ emitters, such as yttrium-90 (2,280 keV) or zirconium-89 (909 keV), instead of fluorine-18 (633 keV) or copper-64 (574 keV) investigated in this study.

Lanthanide-based radioluminescent nanoparticles have also been evaluated as autonomous scintillators using internalized radionuclides as sources of excitation [112,161]. For example, terbium-doped Gd_2_O_2_S (Gd_2_O_2_S:Tb) nanoparticles have been evaluated for use as X-ray nanophosphors; activated in situ by ^18^FDG [162]. However, the biocompatibility of Gd_2_O_2_S:Tb remains a significant concern for further clinical translation towards PDT applications, in addition to the relative short wavelength of its emission (520−550 nm).

Perhaps the greatest impediment, however, to CR/radionuclide activation approaches is the inability to precisely control the delivery of PDT once the photosensitizer was administered. Even with the colocalization of both radionuclide and photosensitizer on the same nanoplatform, once injected such approaches become autonomous by virtue of their physical isolation from the external environment. Dose fractionation, routinely used in radiation therapy to permit reoxygenation of treated, hypoxic tissues, is not an option nor is fine tuning of the radiation dose delivered for photosensitizer activation. As such, many of the operational aspects of internal-source PDT more closely resemble chemotherapy than conventional PDT.

## 7. Conclusions and Future Development

In the preceding sections, we described the underlying principles and evolution of photodynamic therapy: from its initial and still dominant use of light activated, small molecule photosensitizers that passively accumulate in tumors via enhanced permeation, to its latest development of X-ray activated, scintillator–photosensitizer hybrid nanoplatforms that actively target cancer biomarkers. The use of X-rays as the source of photosensitizer excitation enables new paradigms in treatment not previously possible, such as the non-invasive deep tissue treatment of targeted tumors, and precludes many if not most of the therapeutic limitations and side effects of optically driven traditional photosensitizers. The vast and growing number of both X-ray scintillators and photosensitizers offers enormous potential and opportunities for the exploration of synergistic combinations. And, while none of these X-ray activated hybrid nanoplatforms has yet been approved for clinical use, preclinical studies have been highly encouraging.

Prior to clinical use, however, a number of aspects of platform design require further investigation and optimization. Biodistribution, biocompatibility, bioelimination, and long-term/residual toxicity need much greater attention than has been afforded to date. Indeed most in vivo studies performed with the nanocomposites mentioned above have employed clinically irrelevant, highly contrived approaches, such as injecting high concentrations of nanoparticles intratumorally into the xenografts of immunocompromised mice. Preclinical studies that more closely emulate clinical presentations (e.g., spontaneous or humanized murine models, with systemically administered doses of targeted nanoplatforms at translatable concentrations) are sorely needed since typically less than 1% of systemically administered targeted nanoparticles reach their intended destination, due to reticuloendothelial system (RES: liver, spleen) and kidney (particles < 8 nm diameter) clearance [101,163,164]. Precluding protein adsorption (which facilitates RES removal and elicits immune response/sequestering) while enhancing nanoparticle uptake by targeted cells mandate tuning nanoplatform morphology (size, shape, surface charge, and hydrophobicity) and targeting in ways that are relevant to the platform’s route of entry and the tumor’s biology. Tumor topology is rarely homogeneous, often possessing irregular and disorganized vascular structures that lead to heterogenous nanoplatform distribution and treatment. Complicating matters further is the intrinsic toxicity of high-Z materials (e.g., Cd, Se, Te, Pb, Ce, La, Gd, and Hf) frequently incorporated into these nanoplatforms to enhance their X-ray absorption and ROS yield. Bioelimination/accumulation studies of such platforms is rarely conducted but urgently needed, as is the use or more biocompatible high-Z materials such as Bi, I, and Br.

Despite these deficiencies in studies to date, the ability to remotely and controllably activate ROS generation anywhere within the body non-invasively is compelling, as is the diversity of nanoscintillator–photosensitizer materials being developed. Current efforts aimed at enhancing the energy transfer between scintillators and photosensitizers, as well as the development of photosensitizer-free constructs, could soon bring the quantum efficiency of these X-ray-activated nanoplatforms to equal or surpass those of light driven photosensitizers. With further optimization and tailoring of nanoplatform design for clinical application, X-ray PDT could prove invaluable both as a standalone tool and as an adjunct to radiation therapy.

## Figures and Tables

**Figure 1 nanomaterials-13-00673-f001:**
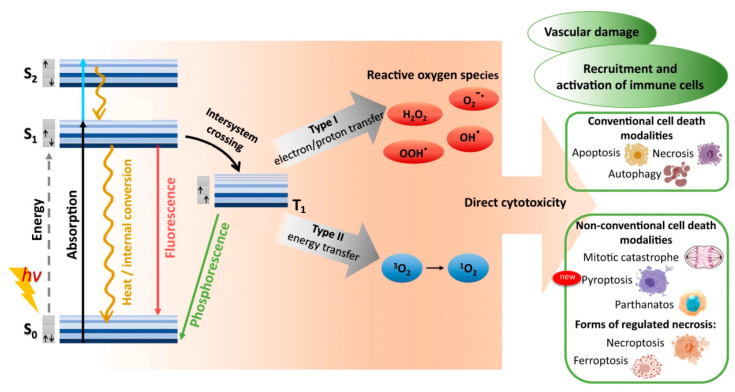
Optical photosensitizer excitation/relaxation, reactive oxygen species production, and cytotoxic outcomes of conventional photodynamic therapy. Absorption of photon energy (hν) can rapidly promote a photosensitizer from its ground singlet state (S_0_) to a non-equilibrium excited singlet state (S_1,2,…n_). Direct excitation to a triplet excited state (T_1,2,…n_) is forbidden, due to conservation of angular momentum. The excited photosensitizer can then lose energy by vibrational relaxation and/or internal conversion, eventually reaching the lowest vibrational level of the excited singlet state (S_1_). Return to the ground singlet state (S_0_) then proceeds either directly via fluorescence or internal conversion/heat, or indirectly via intersystem crossing to the first triplet excited state (T_1_), followed by either phosphorescence or inter/intra-molecular energy transfer that can generate reactive oxygen species. Reproduced and modified with permission [17]. Copyright 2022, Springer Nature.

**Figure 2 nanomaterials-13-00673-f002:**
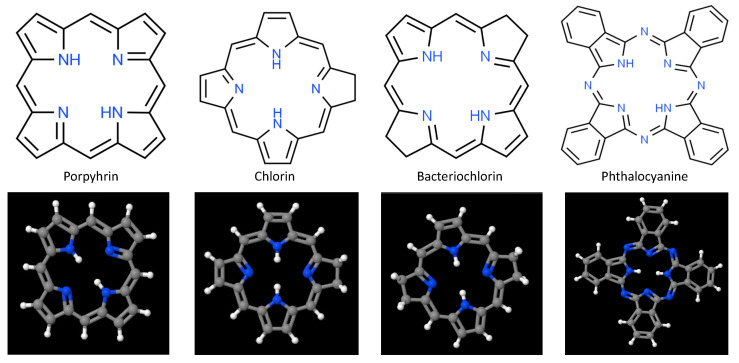
Fundamental chemical structure of porphyrin and porphyrin analogs frequently used as 1st-, 2nd-, and 3rd-generation photosensitizers in photodynamic therapy.

**Figure 3 nanomaterials-13-00673-f003:**
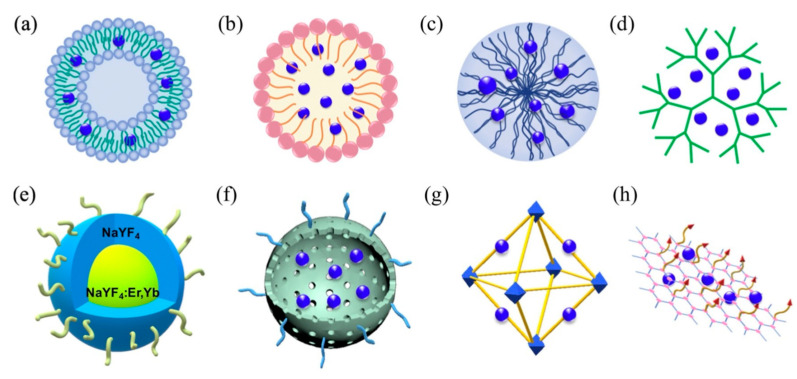
Schematic representation of primary 3rd-generation photosensitizer nanoplatforms: (**a**) liposome, (**b**) micelle, (**c**) polymeric nanoparticles, (**d**) dendrimers, (**e**) upconversion nanoparticles, (**f**) mesoporous silica, (**g**) metal–organic framework, and (**h**) graphene. Reproduced with permission [100]. Copyright 2020, Elsevier.

**Figure 4 nanomaterials-13-00673-f004:**
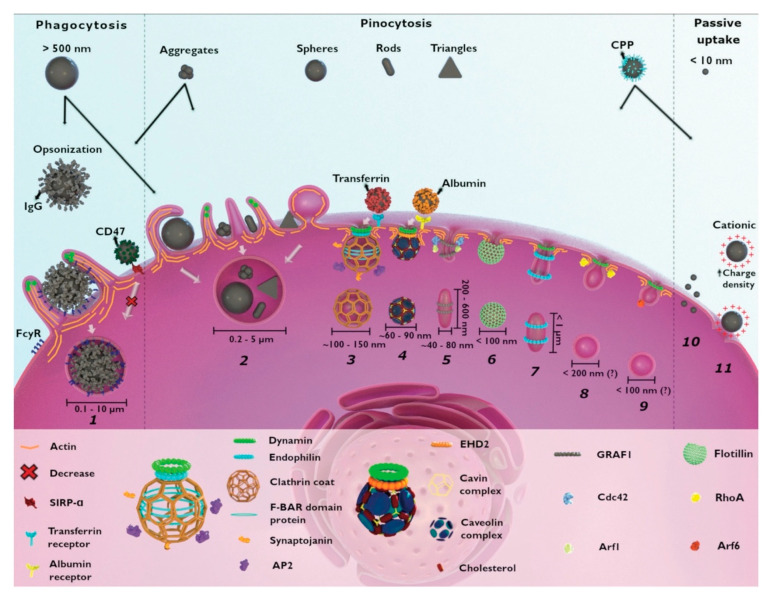
Cellular mechanisms for the uptake of nanoparticles include (1) phagocytosis, (2) macropinocytosis, (3) clathrin-mediated endocytosis, (4) caveolae-dependent endocytosis, (5) CLIC-GEEC, (6) clotillin-mediated endocytosis, (7) fast endophilin-mediated endocytosis, (8) RhoA-dependent endocytosis, and (9) Arf-associated endocytosis. Non-selective macropinocytosis of smaller (<10 nm diameter) and highly charged cationic nanoparticles are internalized via direct penetration (10) and pore formation (11), respectively. Reproduced with permission [105]. Copyright 2021, Royal Society of Chemistry.

**Figure 6 nanomaterials-13-00673-f006:**
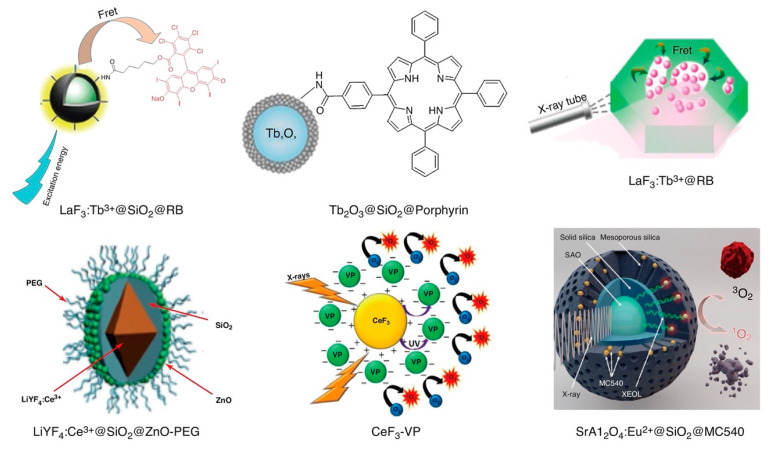
Schematic structural representations of various X-ray PDT radio-photosensitizers based upon rare earth elements: LaF_3_:Tb^3+^@SiO_2_@RB [126], LaF_3_:Tb^3+^@RB [127], CeF_3_-VP [128], LiYF_4_:Ce^3+^@SiO_2_@ZnO-PEG [129], SrAl_2_O_4_:Eu^2+^@SiO_2_@MC540 [130,131], and Tb_2_O_3_@SiO_2_@Porphyrin [132]. Reproduced with permission [123]. Copyright 2018, Elsevier.

**Figure 7 nanomaterials-13-00673-f007:**
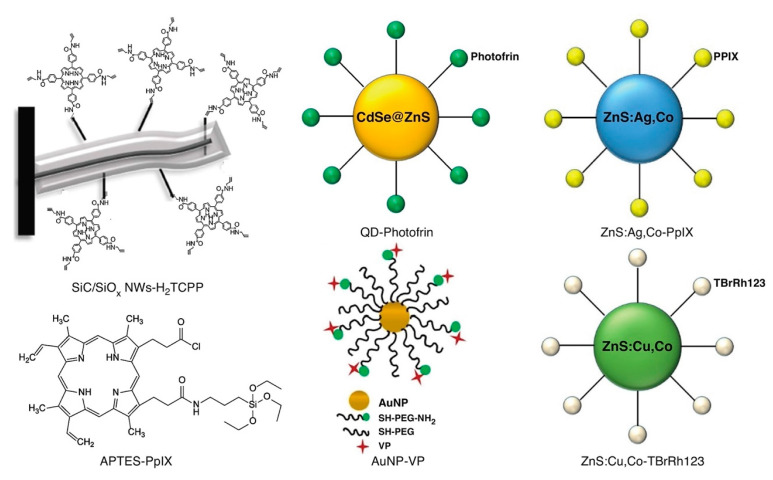
Schematic representation of the structure of various other types of X-ray-induced sensitizers used in X-ray PDT including metal, quantum dot, and silicon: gold nanoparticle (AuNP)-VP [133], APTES-PpIX [134], quantum dot (QD)-Photofrin [135], SiC/SiO_x_ NWs-H_2_TCPP [136], ZnS:Cu,Co-TBrRh123 [137], and ZnS:Ag,Co-PpIX [138]. Reproduced with permission [123]. Copyright 2018, Elsevier.

**Figure 8 nanomaterials-13-00673-f008:**
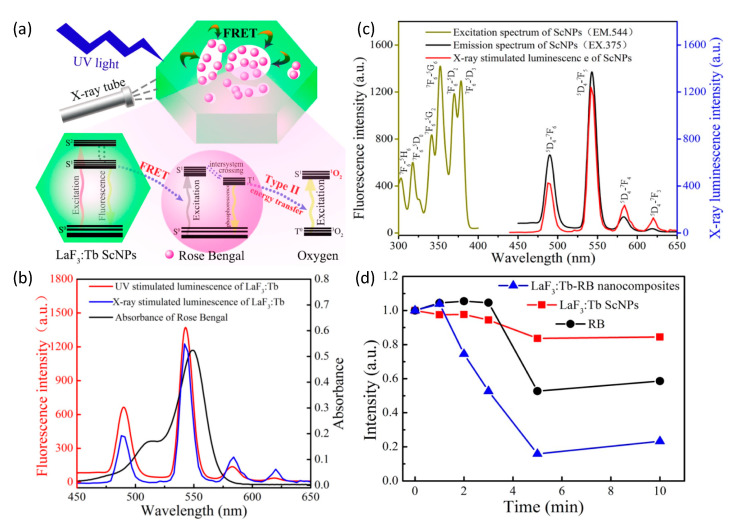
(**a**) Schematic representation of a highly efficient, FRET-driven mesoporous LaF_3_:Tb-RB scintillating nanoparticle for X-ray PDT. (**b**) Nearly identical luminescence spectra of LaF_3_:Tb-RB under X-ray and UV (375 nm) excitation. (**c**) Spectral overlap of LaF_3_:Tb radioluminescence and optical absorption of Rose Bengal. (**d**) Quenching of DPBF fluorescence intensity (inversely proportional to singlet oxygen ^1^O_2_ concentration) as a function of X-ray irradiation exposure, reflecting the importance of FRET in ^1^O_2_ production in the nanocomposite particle. Reproduced with permission [127]. Copyright 2015, American Chemical Society.

**Figure 9 nanomaterials-13-00673-f009:**
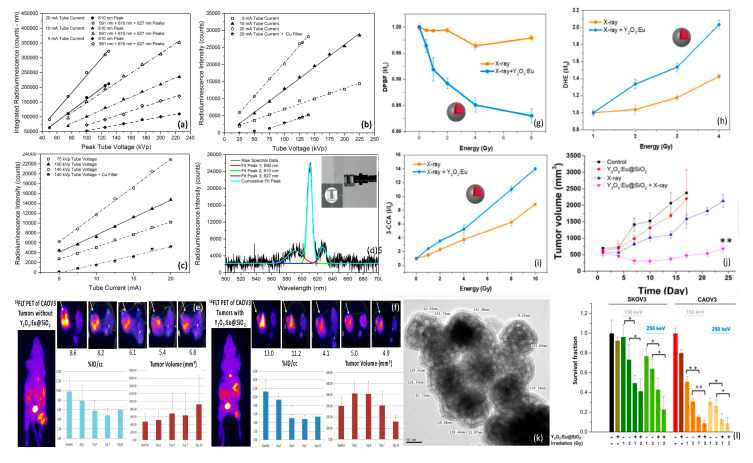
Dosimetric optical response of Y_2_O_3_:Eu@SiO_2_ nanoparticles: (**a**) Area-under-curve of 610 nm and 590 nm + 610 nm + 627 nm radioluminescence peaks as a function of peak tube voltage (proportional to X-ray energy) and tube current (proportional to X-ray flux). (**b**,**c**) Intensity of 610 nm ^5^D_0_ → ^7^F_2_ principal radioluminescence peak, as a function of tube voltage and current. (**d**) Typical X-ray radioluminescence spectra and sampling geometry. (**e**,**f**) ^18^FLT (PET cell proliferation marker) uptake and ovarian cancer tumor volume w/wo Y_2_O_3_:Eu@SiO_2_ nanoparticle X-ray PDT treatment. ROS generation of Y_2_O_3_:Eu@SiO_2_ nanoparticles as a function of absorbed X-ray energy: (**g**) DPBF absorbance reflecting ^1^O_2_, (**h**) DHE emission reflecting O_2_**^−^**, and (**i**) 3-CCA emission reflecting OH. (**j**) In vivo assessment of tumor growth delay afforded by X-ray irradiation of Y_2_O_3_:Eu@SiO_2_ nanoparticles in mice bearing human ovarian cancer xenografts w/wo 8 Gy X-ray irradiation. (**k**) TEM showing particle morphology and SiO_2_ encasement. (**l**) Clonogenic assay results using radiation sensitive and resistant human ovarian cancer cells. Figures (**a**–**d**) reproduced with permission [144]. Copyright 2014, American Institute of Physics. Figures (**g**–**j**,**l**) reproduced with permission [109]. Copyright 2020, Ivyspring International.

**Figure 10 nanomaterials-13-00673-f010:**
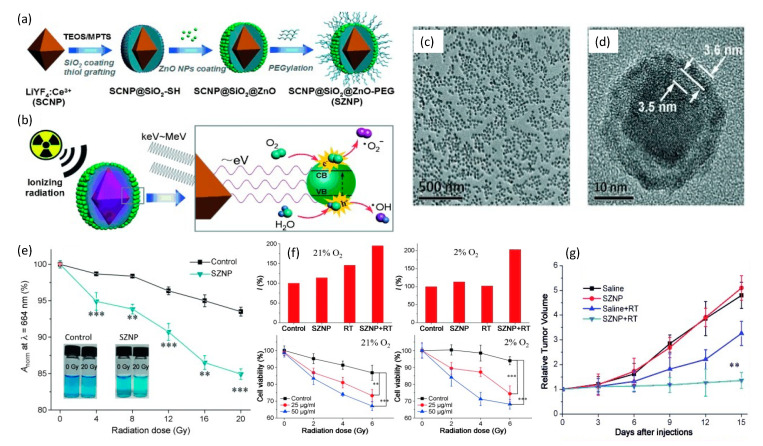
(**a**) Construction and (**b**) mechanism of action of a core–shell of LiYF_4_:Ce^3+^@SiO_2_@ZnO-PEG nanoparticle. (**c**,**d**) TEM images reflecting both high degree of polydispersity and internal structure of the nanoplatform. (**e**) Comparison of ROS production between control and SZNP as a function of radiation dose via methylene blue abruption at 664nm. (**f**) Hypoxix/normoxic HeLa cell studies showing significant decrease in cell viability due to SZNP X-ray irradiation. (**g**) Murine studies reflecting change in relative HeLa tumor volume as a function of time for different treatments with 8 Gy dose. Reproduced with permission [129]. Copyright 2015, John Wiley & Sons.

**Figure 11 nanomaterials-13-00673-f011:**
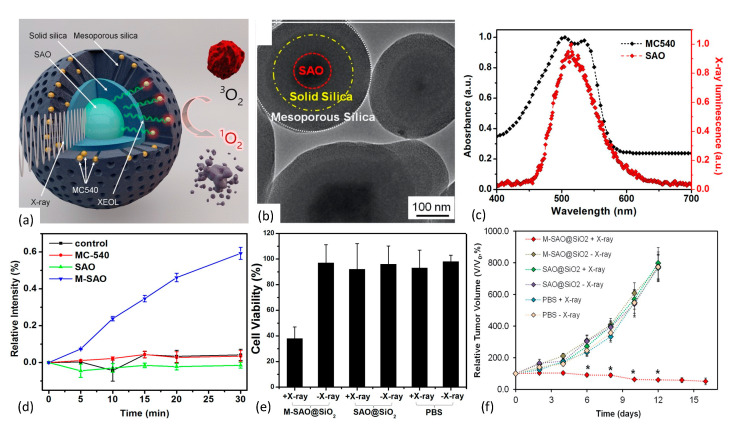
(**a**,**b**) Nanoscintillator core fabricated of SrAl_2_O_4_:Eu^2+^ is enveloped within two layers of silica, an inner solid layer and an outer mesoporous layer, with the latter’s pores loaded with the photosensitizer MC540. Under X-ray irradiation, SAO converts X-rays to visible light that, in turn, activates nearby MC540 molecules to produce cytotoxic ^1^O_2_. (**b**) TEM showing core/dual-shell structure. (**c**) Spectral overlap of nanoscintillator emission and photosensitizer absorption bands. (**d**) Singlet oxygen production as measured via SOSG assay. (**e**) MTT assay results showing impact of X-ray irradiated nanoplatform on U87MG cell survivability. (**f**) Tumor growth curves reflecting impact of ^1^O_2_ production by M-SAO@SiO_2_ nanoparticles. Reproduced with permission [130]. Copyright 2015, American Chemical Society.

**Figure 12 nanomaterials-13-00673-f012:**
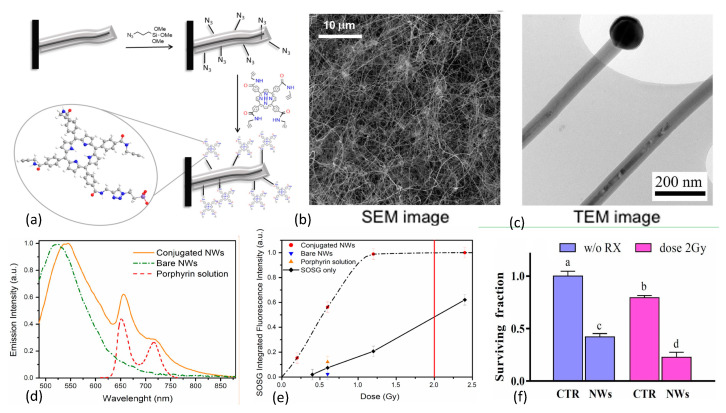
(**a**) Design and conjugation of porphyrin to azide functionalized SiC/SiO_x_ nanowires (NWs). (**b**,**c**) TEM and SEM images of NW network on Si substrate. (**d**) Fluorescence emission spectra of conjugated/bare NWs and porphyrin. (**e**) ^1^O_2_ production as a function of X-ray dose for bare and porphyrin conjugated NWs in water with controls, as measured via SOSG marker. Standard clinical 2 Gy dose denoted by red vertical line. (**f**) Results of clonogenic survival assays of A549 adenocarcinoma alveolar basal epithelial cells using 6 MV X-rays at 2 Gy doses, demonstrating that 12 days after single irradiation, cell populations were reduced by about 75% with respect to controls. Reproduced with permission [136]. Copyright 2015, Springer Nature.

**Figure 13 nanomaterials-13-00673-f013:**
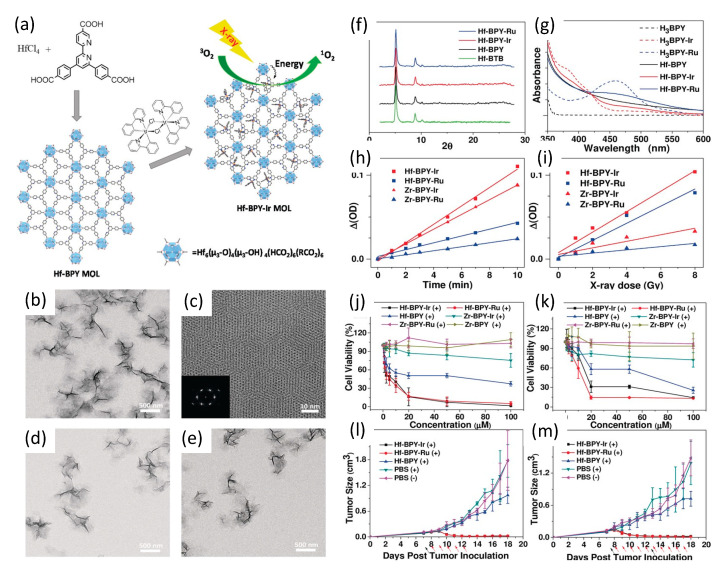
(**a**) Synthesis of Hf-based MOLs and MOL-enabled X-ray PDT to generate ^1^O_2_. (**b**) TEM image. (**c**) HRTEM image (with FFT pattern insert of Hf-BPY. TEM images of (**d**) Hf-BPY-Ir, and (**e**) Hf-BPY-Ru. (**f**) Powder X-ray diffraction (PXRD) patterns of Hf-BPY-Ir, Hf-BPY-Ru, and Hf-BPY. (**g**) UV/Vis absorption spectra of Hf-MOLs and bridging ligands. Singlet oxygen generation of Hf- and Zr-MOLs upon visible light irradiation (**h**) or X-ray irradiation (**i**). In vitro and in vivo anticancer efficacy of Hf-MOLs. Cytotoxicity of Hf- and Zr- MOLs in CT26 cells (**j**) and MC38 cells (**k**). Tumor growth inhibition curves after X-PDT treatment in the CT26 (**l**) and MC38 (**m**) murine models. Black arrows indicate dates of injection of MOLs; red arrows indicate dates of X-ray irradiation. Reproduced with permission [155]. Copyright 2017, John Wiley & Sons.

**Figure 14 nanomaterials-13-00673-f014:**
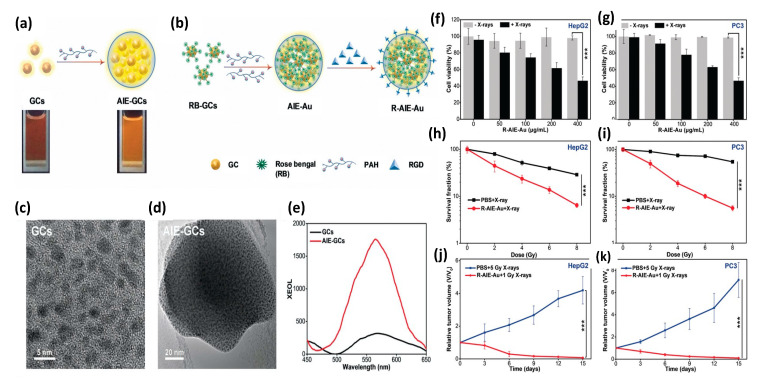
(**a**) Schematics of AIE-GCs and (**b**) R-AIE-Au nanosensitizers. TEM images of (**c**) GCs and (**d**) AIE-GCs. (**e**) Luminescence spectra of GCs and AIE-GCs activated by X-ray irradiation at 50 kV_p_ tube potential and 70 μA tube current. Viability of (**f**) HepG2 and (**g**) PC3 cells incubated with various concentrations of R-AIE-Au w/wo X-ray irradiation. Clonogenic assays of (**h**) HepG2 and (**i**) PC3 cells 10 days following X-ray irradiation, w/wo R-AIE-Au. Tumor growth curves for (**j**) HepG2 and (**k**) PC3 tumors in mice subjected to various treatments. Reproduced with permission [156]. Copyright 2020, John Wiley & Sons.

## Data Availability

Not applicable.

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
