# Peer review of "X-ray Activated Nanoplatforms for Deep Tissue Photodynamic Therapy"

_nanomaterials, 2023, doi:10.3390/nano13040673_

Round 1
Reviewer 1 Report
This manuscript details the principles and evolution of photodynamic therapy starting from that which makes use of small molecule photosensitizers that are activated by light, to the recent development of X-ray activated hybrid scintillator-photosensitizer nanoplatforms that actively target biomarkers of the cancer. These nanoplatforms produce large amounts of ROS when irradiated with diagnostic x-ray doses and energies, and enable photodynamic therapy of deeper tissues without surgery or laparoscopy to reach those organs. None of the nanoplatforms have entered clinical use yet because further investigations are still needed to evaluate carefully. biodistribution, biocompatibility, bioelimination and possible toxicity. Although there are still many studies to be done to optimize and customize the nanoplatform design for clinical application, this paper reports and clearly demonstrates that X-ray PDT could prove valuable for anticancer therapy both in association with radiotherapy and independently.
The work is certainly interesting and the subject can stimulate the attention of a large number of readers.
Author Response
We deeply appreciate your in-depth review of our manuscript and kind words of support. We too hope our paper stimulates broader interest in, and appreciation of, this new and potentially paradigm shifting form of photodynamic therapy.
Reviewer 2 Report
In this manuscript, “X-ray Activated Nanoplatforms for Deep Tissue Photodynamic Therapy” by Souris et al. examined the underlying principles and evolution of PDT: from its initial use of light activated, small molecule photosensitizers that passively accumulate in tumors, to its latest development of x-ray activated, scintillator-photosensitizer hybrid nanoplatforms that actively target cancer biomarkers. This work is well written and could be accepted in its current form.
Author Response
We deeply appreciate your in-depth review of our manuscript and kind words of support. We hope our paper stimulates broader interest in, and appreciation of, this new and potentially paradigm shifting form of photodynamic therapy.
Reviewer 3 Report
Very nice, very well written, a huge amount of data and references , I don't have anything else to say, except to add another Energy Down Converting Process Cascade, called XPACT (X-ray psoeral Activated Cancer Therapy, that works in the same way: the photosensitizer is a psoralen derivative. The mechanism: the X-ray exciting an Energy Modulator – rare earth oxide; Energy Modulator absorbs X-Ray and emits UVA radiation; in situ produced UVA excites co-administered psoralen (8-MOP), Psoralen activation generates anti-tumor response, resulting in cell apoptotic death.
Author Response
We deeply appreciate your in-depth review of our manuscript and kind words of support. We truly hope our paper stimulates broader interest in, and appreciation of, this new and potentially paradigm shifting form of photodynamic therapy.
We also thank you for pointing out our accidental omission of X-PACT and have, accordingly, modified our manuscript at lines 546-553 (and added 2 references that are, temporarily, appended to the References section, pending editorial staff insertion/renumbering at line 553) to read:
"In some X-ray PDT implementations, scintillators and photosensitizers are delivered independently of one another. An example of this approach, currently undergoing Phase I clinical trial (NCT04389281), is X-ray Psoralen Activated Cancer Therapy (X-PACT). In this technique, x-ray/UV down-converting rare-earth nanoscintillators like Y2O3 or nanophosphors like CaWO4 are co-administered with UV-activated psoralen-derivative photosensitizers like 8-Methoxypsoralen (8-MOP), to yield deep tissue apoptosis upon x-ray irradiation [1, 2]."
- Oldham, M.; Yoon, P.; Fathi, Z.; Beyer, W. F.; Adamson, J.; Liu, L.; Alcorta, D.; Xia, W.; Osada, T.; Liu, C.; et al. X-Ray Psoralen Activated Cancer Therapy (X-PACT). PLoS One 2016, 11 (9), e0162078. DOI: 10.1371/journal.pone.0162078.
- Scaffidi, J. P.; Gregas, M. K.; Lauly, B.; Zhang, Y.; Vo-Dinh, T. Activity of psoralen-functionalized nanoscintillators against cancer cells upon X-ray excitation. ACS Nano 2011, 5 (6), 4679-4687. DOI: 10.1021/nn200511m.
Reviewer 4 Report
Review Report on Dec20, 2022
X-ray Activated Nanoplatforms for Deep Tissue Photodynamic Therapy
This is an interesting review and the authors have collected a unique dataset using innovative methodology on Photodynamic Therapy. The review is well written and structured. However, in my opinion, the review has some shortcomings in regards to some data analyses and text, and I feel this unique dataset has not been utilized fully. Below, I have provided numerous remarks on the text, as it is often unclear and long-winded. Furthermore, I made additional suggestions for more in-depth analyses of the review. Given these shortcomings, the manuscript requires minor revisions.
Key points to be answered:
1. Authors should provide the excited states in the Fig 1. The Fig. 1 is unclear to understand the mechanism of transition states from Femto second (10-15) to nano seconds, compared to the Sec. 2.1 explanations.
2. Fig.2. Porphin structure is not correct as the two pyrrole rings should be opposite but your structure is not same. Porphin is composed of four modified pyrrole subunits interconnected at their α carbon atoms via methine bridges (=CH−). Rectify the structure. In addition, authors should follow same nomenclature on structure and write up such as either Porphin or Porphyrin.
3. Line 362-363 should be corrected.
4. In Fig. 5, (line 450), the authors have mentioned “Relaxation mechanisms and density of states for a simplified scintillator (insulator): I am confused with them, is it insulator or fluorescene emitter?
“As I know is that Scintillator is a general term for substances that emit fluorescence when exposed to radiation such as X-rays and γ-rays, it is a type of phosphor.”
5. What is FRET (line 579), there is no full form in the chapter.
6. Figure 9 is not clear to understand, needs tobe changed of all font sizes.
The overall manuscript is well written on establishment of research data X-ray Activated Nanoplatforms for Deep Tissue PhotodynamicTherapy. I found typos in each paragraph, and correct them. The images are in poor quality, many data points are not visible in the graphs. This re view article is suitable for publication after major review.
